# ExBEHRT: Extended Transformer for Electronic Health Records

**Maurice Rupp**
Novartis Oncology AG
Basel, Switzerland
`maurice.rupp@gmail.com`

**Oriane Peter**
Novartis Oncology AG
Basel, Switzerland
`oriane.peter@gmail.com`

**Thirupathi Pattipaka**
Novartis Oncology AG
Basel, Switzerland
`thirupathi.pattipaka@novartis.com`

## Abstract

In this study, we introduce ExBEHRT, an extended version of BEHRT (BERT applied to electronic health record data) and applied various algorithms to interpret its results. While BEHRT only considers diagnoses and patient age, we extend the feature space to several multi-modal records, namely demographics, clinical characteristics, vital signs, smoking status, diagnoses, procedures, medications and lab tests by applying a novel method to unify the frequencies and temporal dimensions of the different features. We show that additional features significantly improve model performance for various down-stream tasks in different diseases. To ensure robustness, we interpret the model predictions using an adaption of expected gradients, which has not been applied to transformers with EHR data so far and provides more granular interpretations than previous approaches such as feature and token importances. Furthermore, by clustering the models' representations of oncology patients, we show that the model has implicit understanding of the disease and is able to classify patients with same cancer type into different risk groups. Given the additional features and interpretability, ExBEHRT can help making informed decisions about disease progressions, diagnoses and risk factors of various diseases.

## 1 Introduction

Over the last decade, electronic health records have become increasingly popular to document a patient's treatments, lab results, vital signs, etc. Commonly, a sequence of medical events is referred to as a *patient journey*. Given the immense amount of longitudinal data available, there lies tremendous potential for machine learning to provide novel insights about the recognition of disease patterns, progression and subgroups as well as treatment planning. Recent studies have adapted transformers to structured tabular EHR data and demonstrated their superiority in various benchmarks compared to other similar algorithms (Kalyan et al., 2022). Although there exists work on unstructured freetext EHR data (e.g. BioBERT (Lee et al., 2019)), these models are out of the scope of this study.

## 2 Related Work

The first adaptation of transformers to structured EHR data, called BEHRT (Li et al., 2020), incorporated diagnosis concepts and age from EHRs and added embeddings for the separation of individual visits and a positional embedding for the visit number. Other models such as Med-BERT (Rasmy et al., 2021), CEHR-BERT (Pang et al., 2021) and BRLTM (Meng et al., 2021) added more features by concatenating the inputs into one long patient sequence. These approaches are limited in the amount of data from a single patient they can process and the computational power required increases significantly with each feature added. In addition, there exists a variety of models that either

combine the BERT architecture with other machine learning models (Shang et al. (2019), Poulain et al. (2022), Li et al. (2021)) or focus only on disease-specific use-cases (Azhir et al. (2022), Prakash et al. (2021), Rao et al. (2022)). These models lack generalizability to other tasks due to their unique training methodologies and domains.

In this work, we present a novel approach to integrate multimodal features into transformer models by adding medical concepts separately and vertically instead of chaining all concepts horizontally. We show that these features are important in various downstream applications such as mortality prediction, patient subtyping and disease progression prediction.

## 3    EXBEHRT FOR EHR REPRESENTATION LEARNING

ExBEHRT is an extension of BEHRT where medical concepts are not concatenated into one long vector, but grouped into separate, learnable embeddings per concept type. In this way, we avoid exploding input lengths when adding new medical features and give the model the opportunity to learn which concepts it should focus on. From a clinical perspective, it would also be useful to separate diagnoses, procedures, drugs, etc., as they have different clinical value for downstream applications. We take the number of diagnoses in a visit as an indicator of how many "horizontal slots" are available for other concepts in that visit (e.g. two for the first visit in figure 1). Therefore, the maximum length of the patient journey is defined by the number of diagnosis codes of a patient, regardless of the number of other concepts added to the model. Another advantage of this procedure is that it can handle different frequencies and small amounts of additional concepts. As shown by the procedures in figure 1, but carried out in the same way with lab tests, there are three possible cases of adding a new concept to a visit:

1. The number of procedures is equal to the amount of horizontal slots available in the visit (visit 1 - two each). The procedures can therefore be represented as a 1D vector.

2. The number of procedures exceeds the amount of slots available in the visit (visit 2 - one diagnosis, two procedures). Here, the procedures fill up the number of horizontal slots line by line until there are no more procedures left, resulting in a 2D vector of dimensions $\#slots \times \lceil \frac{\#procedures}{\#slots} \rceil$.

3. The number of procedures subceeds the amount of slots available (visit 3 - one diagnosis, no procedures). The procedures are represented as a 1D vector and then padded to the amount of horizontal slots available.

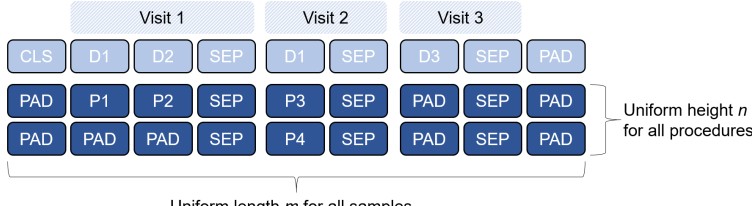

Figure 1: An example of how ExBEHRT represents a patient with a constant sentence length $m$.

After reshaping, all procedures and labs of all patients are padded to the same amount of rows $n$ to enable batch processing. Before passing the inputs to the model, each token is embedded into a 288-dimensional vector and all tokens are summed vertically. Figure 7 in the appendix shows the final representation of one patient.

### 3.1    DATA

In this study, we used the Optum® de-identified EHR database. It is derived from healthcare provider organizations in the United States, which include more than 57 contributing sources and 111,000 sites of care including hospital-based medical services networks comprising academic, private, and community hospitals treating more than 106 million patients. Optum® data elements

also include demographics, medications prescribed and administered, immunizations, allergies, lab results (including microbiology), vital signs and other observable measurements, clinical and hospitalisation administrative data, and coded diagnoses and procedures. The population in Optum® EHR is geographically diverse, spanning all 50US states. We selected only data points collected during hospitalisations to ensure data quality and consistency. Each patient must have at least five visits with valid ICD-9 or ICD-10 diagnosis codes to ensure sufficient temporal context. Considering these criteria, our final pre-training cohort consisted of 5.4 million individual patients divided into training (80%), validation (10%) and testing (10%) groups. Table 1 shows the characteristics of the final cohort.

Table 1: Statistics of the cohort used for pre-training.

| Feature | Metric |
|---|---|
| Birth year | 1973±25, min: 1932, max: 2021 |
| Gender | 41.49% male, 58.51% female |
| Distribution by race | 68% Cau., 22% Afr. Am., 1% As., 9% other |
| No. of diagnosis codes per patient | 14±11.1, min: 5, max: 121 |
| No. of visits per patient | 9±6.6, min: 5, max: 63 |
| % of patients without labs | 14.33% |
| % of patients without procedures | 1.64% |
| % of patients without BMI | 21.74% |
| % of patients without smoking status | 27.11% |
| % of deceased patients | 14.52% |

## 3.2 MODEL TRAINING

ExBEHRT consists of the same model architecture as BEHRT. For pre-training, we applied the standard MLM procedure described in the original BERT paper (Devlin et al., 2018) for predicting masked diagnosis codes using their BertAdam optimizer with cross-entropy loss. All BERT-based models (ExBEHRT, BEHRT, Med-BERT[1]) were trained for 40 epochs on one Tesla T4 GPU with 16GB memory, selecting the epoch with the highest micro-averaged MLM precision score on the validation set. To ensure a fair comparison, we used the same amount of attention layers (6) and heads (12) as well as embedding dimension (288) for all three BERT-based models. We also pre-trained a version of ExBEHRT on the additional pre-training objective PLOS[2] as introduced by Med-BERT, which we called ExBEHRT+P.

In a second step, we fine-tuned the models on two prediction tasks: Death of a patient within six months after the first cancer diagnosis and readmission into hospital within 30 or fewer days after heart failure. All tokens after the cancer diagnosis/heart failure are not disclosed to the model. The cohorts for these two tasks were split into 80% training, and 10% each validation and testing datasets, with each patient present in both cohorts (pre-training and fine-tuning) assigned to the same split for both tasks. The cohorts consist of 437'902 patients (31.67% deceased within 6 months after first cancer diagnosis) for *Death in 6M* and 503'161 patients (28.24% readmitted within 30 days) for *HF readmit*. Furthermore, we used the patient representations of ExBEHRT to identify risk subtypes of cancer patients using unsupervised clustering. For this purpose, we used a combination of the dimensionality reduction technique UMAP (McInnes et al., 2018) and the clustering algorithm HDBSCAN (Campello et al., 2013) and applied the clustering to all cancer patients from the pre-training cohort (260'645).

---

[1]We used the public code from both publications and adjusted the models to work with our data structures: https://github.com/deepmedicine/BEHRT and https://github.com/ZhiGroup/Med-BERT

[2]Binary classification of whether a patient had at least one prolonged length of stay in hospital (> 7 days) during their journey.

## 4 RESULTS

### 4.1 EVENT PREDICTION

For each experiment, we selected the model with the best validation precision score and only then evaluated the performance on the test set. The metrics used for evaluation are the area under the receiver operating characteristic curve (AUROC), average precision score (APS) as well as the precision at the 0.5 threshold. In all but one metric in one task, ExBEHRT outperforms BEHRT, Med-BERT and other conventional algorithms such as Logistic Regression (LR) and XGBoost when evaluated on this hold-out dataset.

Table 2: Average fine-tuning results of various models and their standard deviations.

| Task | Metric | LR | XGB | BEHRT | Med-BERT | ExBEHRT | ExBEHRT+P |
|------|--------|-----|-----|-------|----------|---------|-----------|
| | APS | 42.8±0.0% | 45.5±0.1% | 47.7±0.4% | 46.2±0.4% | **53.1±0.3%** | 52.6±0.3% |
| Death in 6M | AUROC | 63.5±0.0% | 66.4±0.1% | 66.7±0.6% | 65.3±0.3% | **71.5±0.5%** | 70.9±0.5% |
| | Precision | 73.0±0.1% | 74.3±0.1% | 75.2±0.2% | 74.5±0.1% | **78.1±0.1%** | 77.9±0.1% |
| | APS | 51.6±0.0% | 45.5±0.1% | 55.5±0.1% | 54.4±0.2% | **59.8±0.2%** | 59.6±0.2% |
| Death in 12M | AUROC | 66.7±0.0% | 66.3±0.1% | 70.1±0.2% | 68.9±0.3% | **74.3±0.4%** | 73.8±0.4% |
| | Precision | 70.4±0.1% | 74.4±0.1% | 73.2±0.1% | 72.4±0.1% | **76.4±0.1%** | 76.3±0.1% |
| | APS | 29.8±0.0% | **31.3±0.1%** | 19.9±0.1% | 19.8±0.1% | 30.0±1.6% | 25.1±0.1% |
| HF readmit | AUROC | 51.9±0.1% | 53.6±0.1% | 51.2±0.1% | 51.0±0.1% | 56.7±1.7% | **56.8±0.2%** |
| | Precision | 72.0±0.0% | 72.3±0.1% | 81.0±0.1% | 81.0±0.0% | 78.7±0.2% | **81.6±0.1%** |

### 4.2 INTERPRETABILITY ON EVENT PREDICTION RESULTS

For all interpretability experiments, we used the ExBEHRT model fine-tuned on the task *Death in 6M*, meaning whether a cancer patient will decease within six months after their first cancer diagnosis. We visualize the interpretability for individual patients only, as both interpretability approaches are example-based and not model-agnostic.

#### 4.2.1 SELF-ATTENTION VISUALIZATION

Analogous to previous work (Li et al. (2020), Rasmy et al. (2021), Meng et al. (2021)), we visualised the attention of the last network layer using BertViz (Vig (2019)). However, since in all such models all embeddings are summed before being passed through the network, self-attention has no way of assigning individual input features to the outcome. Nevertheless, we can draw conclusions about how the different slots interact with each other and which connections the model considers important. Figure 2 shows the self-attention of a single patient in the last layer of ExBEHRT. The left figure shows the attentions of all 12 attention heads in this layer, while the right figure shows the attention of one head. As expected, the model focuses strongly on the slots within a visit, as these slots are by definition strongly interconnected. Slot 7 corresponds to the slot in which the patient was diagnosed with lung cancer. Although the model was not specifically trained on cancer codes, it pays close attention to this slot, indicating that it has learned some correlation between the cancer diagnosis and the predicted outcome. Interestingly, slot 7 receives a lot of attention on the first and second visits, but not on the two previous visits, suggesting that the model is able to learn causality over long time intervals.

#### 4.2.2 EXPECTED GRADIENTS INTERPRETABILITY

Due to the limitations of self-attention visualization, we explored the technique expected gradients (Erion et al., 2020) for deeper understanding of the model. Expected gradients is considered to be one of the most robust gradient-based feature attribution methods for deep learning models. In this way, we can infer the importance of individual features and tokens for predictions, which is not possible with Self-Attention. Since each individual concept (diagnosis code, procedure code, age, etc.) is mapped to a 288-dimensional embedding before being passed to the model, we first calculated the expected gradients for each of the 288 positions and then summed the absolute values to obtain a single gradient value for each input token. In this way, each input token has an associated gradient that is linked to the output of the model and yields detailed insight into its impact on the prediction of the model. We visualised the results in three levels of abstraction with increasing levels of detail in figures 3, 4 and 5.

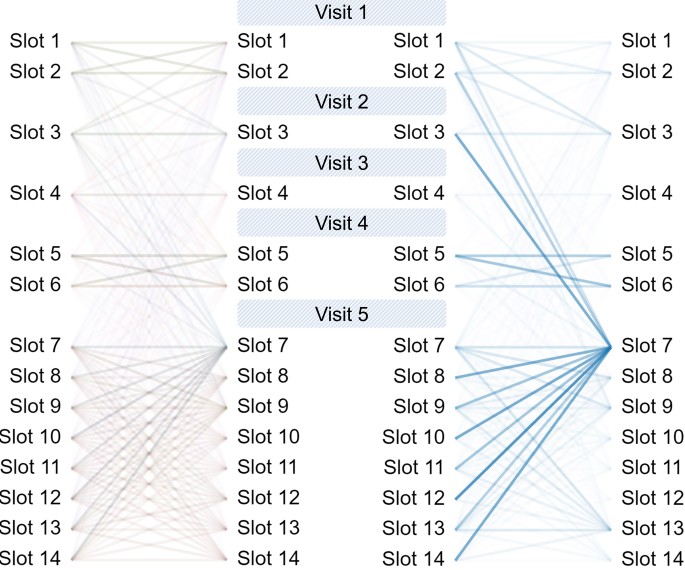

Figure 2: Left: The self-attention of all 12 attention heads of the last layer of ExBEHRT. Higher opacity corresponds to higher attention. Right: The self-attention of one attention head of the last layer. Slot 7 corresponds to the slot where the cancer was diagnosed.

For figure 3, we summed the expected gradients for each of the input features. This allows us to assess the different effects of the features on the outcomes for a particular patient. For this patient, the diagnoses and procedures (treatments & medications) were by far the most important features. With this visualisation, we can also assess basic biases. For example, gender was not considered an important characteristic, suggesting that performance would be similar for a person with a different gender.

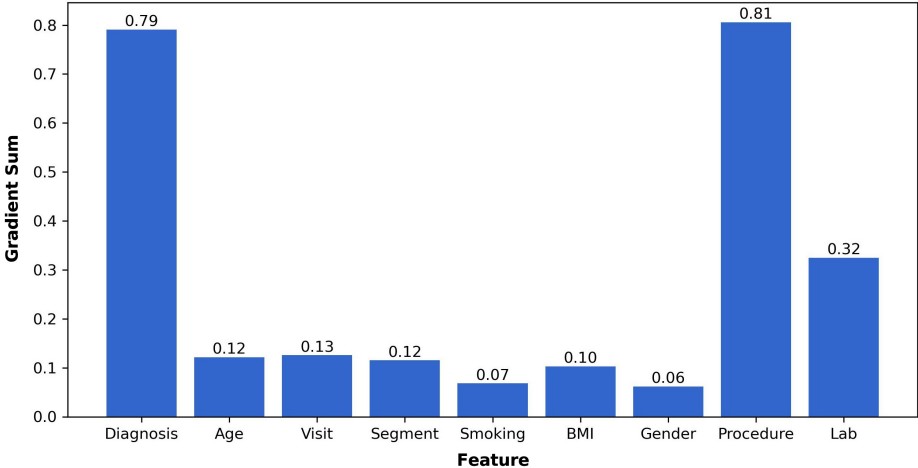

Figure 3: The absolute sums of the expected gradients summed by input feature.

For figure 4, we visualised the absolute expected gradients for each of the input features and summed them at each time slot. In this way, we can assess the importance of the different features over time to get an idea of where the model is focusing. Interestingly, for the first two visits, the model placed more emphasis on what kind of medications and treatments the patient received, while for the last visit (the visit where the patient was diagnosed with blood cancer), it placed more emphasis on diagnoses and labs. In general, slot 5, where the cancer was diagnosed, was given the most importance.

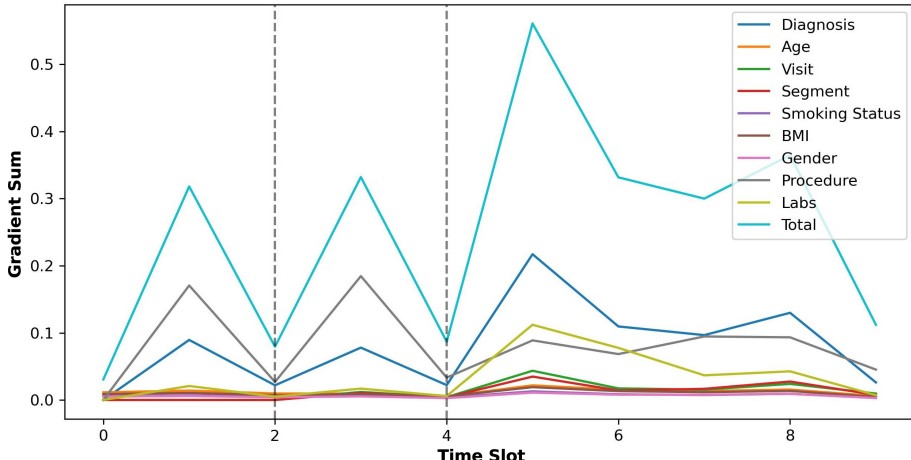

Figure 4: The absolute sums of the expected gradients summed by input feature and time slot. The dotted lines represent slots with SEP tokens and therefore indicate the next visit.

Figure 5 shows the absolute sums of the gradients of each input token and provides a detailed interpretation of which medical concept had which influence on the prediction of the model. Unsurprisingly, the cancer code C81 had the greatest influence on the result. However, earlier codes such as J40 or 71020 also contribute to the model's prediction, indicating that the model is able to incorporate information from the entire patient journey into its results.

| Diagnosis | | | | | | | | | |
|---|---|---|---|---|---|---|---|---|---|
| CLS | J40 | SEP | M54 | SEP | C81 | R55 | R59 | E87 | SEP |
| **Lab** | | | | | | | | | |
| - | - | - | - | - | CHEMISTRY | URINALYSIS | HEMATOLOGY | SPEC. CHEM. | - |
| - | - | - | - | - | SPEC. LAB | BLOOD GAS | - | - | - |
| **Procedure** | | | | | | | | | |
| - | 71020 | - | 81003 | - | - | - | - | - | - |
| - | 94640 | - | 87077 | - | - | - | - | - | - |
| - | 99283 | - | 87086 | - | - | - | - | - | - |

Figure 5: A visualization of the absolute sums of the expected gradients of diagnoses, labs and procedures on a concept level. Darker colours represent higher values and the SEP tokens indicate the separation between two visits.

## 4.3 CANCER PATIENT CLUSTERING

HDBSCAN was able to cluster 90% of all cancer patients into 24 clusters[3]. On average, the most occurring cancer diagnosis within a cluster was present for 84% of the patients assigned to this specific cluster and the mean cluster purity[4] was 85%. Similar concepts (e.g. cancer of female reproductive organs or different types of leukaemia) lay in areas close to each other, indicating a spatial logic between the cancer types. In figure 6, we show that with a second pass of HDBSCAN on a given cluster, we can identify risk subgroups. In all three identified clusters, more than 90% of the patients actually do have pancreatic cancer and all clusters share similar general characteristics.

However, as shown in the table 4.3, ExBEHRT identified a subgroup with a significantly higher chance of recovering from cancer and a lower probability of dying, although this information was not provided to the model at any point in time[5].

---

[3]A visualization of all clusters can be found in figure 8 in the appendix.

[4]Cluster purity indicates the fraction of patients with a condition which are assigned to the cluster.

[5]In the table, *% of journey with cancer* indicates the ratio of the time between the first and last cancer diagnosis compared to the duration of the whole recorded patient journey. *Cancer-free* refers to the percentage of patients within a cluster, which have records of at least two visits without cancer diagnosis after the last visit

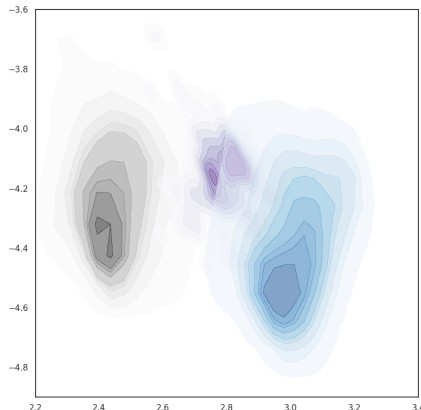

Figure 6: The three identified patient subclusters with pancreatic cancer visualized with a kernel density estimate plot for visual clarity.

Table 3: Statistics of the three pancreatic cancer clusters indicating a clear differentiation between higher risk (grey, blue) and lower risk patients (purple).

| Metric | Gray | Blue | Purple |
|---|---|---|---|
| Median age | 67 | 68 | 68 |
| Median birth year | 1950 | 1947 | 1944 |
| Median BMI | 25 | 25 | 26 |
| Average death rate | 76.5% | 75.9% | **70.0%** |
| % of journey with cancer | 27.0% | 24.0% | **18.3%** |
| Cancer-free | 34.0% | 36.9% | **62.7%** |

## 5 CONCLUSION

In this study, we presented a novel method for adding patient features to BEHRT that significantly increases the predictive power for multiple downstream tasks in different disease domains. The novel method of stacking features vertically yielded improvements in hardware requirements and benchmarks and and facilitates the extension to new concepts in the future. Given the large number and heterogeneity of patients with which the model has been pre-trained, we are confident that ExBEHRT will generalise well to new data, patients and tasks. Combined with interpretability, the model provides more detailed insights into disease trajectories and subtypes of different patients than previous approaches, and could help clinicians form more detailed assessments of their patients' health. Furthermore, with a personalised understanding of patient groups, it is possible to identify unmet needs and improve patient outcomes.

**Limitations**
Nevertheless, there are some limitations: It is extremely difficult to validate the quality, completeness and correctness of EHR datasets, as the data is usually processed anonymously and comes from a variety of heterogeneous, fragmented sources. The sheer nature of EHR data also introduces bias, as physicians may have an incentive to diagnose additional less relevant conditions, as medical billing is closely related to the number and type of diagnoses reported.

**Future Work**
In a potential next step, we would like to verify the results and interpretations of this work with clinicians to ensure robust and sound predictions as possible given the acquired interpretability. In addition, we would like to test the generalisability of ExBEHRT to other clinical use-cases such as severity prediction and risktyping of other diseases as well as specific cancers.

---

with a cancer diagnosis. The *average death rate* comes directly from the EHR database and unfortunately does not include information on the cause of death.

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

# A   APPENDIX

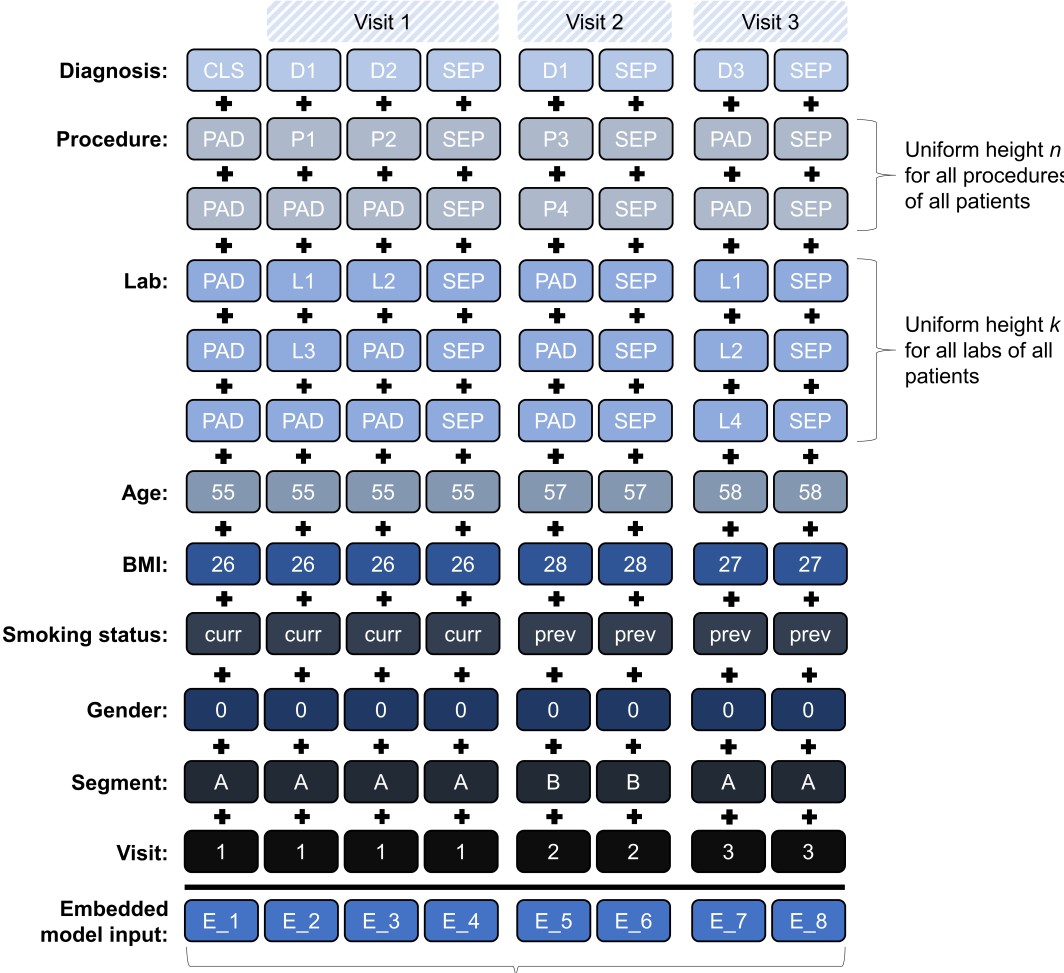

Figure 7: A sample input of ExBEHRT. Each of the concepts (diagnosis, procedure, lab, age, BMI, smoking status, gender, segment, visit) has its own embedding, where each of the tokens is mapped to a 288-dimensional vector, which is learned during model training. After embedding, all concepts are summed vertically element-wise to create a single $288 \times m$ dimensional vector as input for the model. The parameters $n$ and $k$ are set before training and fixed for each patient to ensure coherent batch processing.

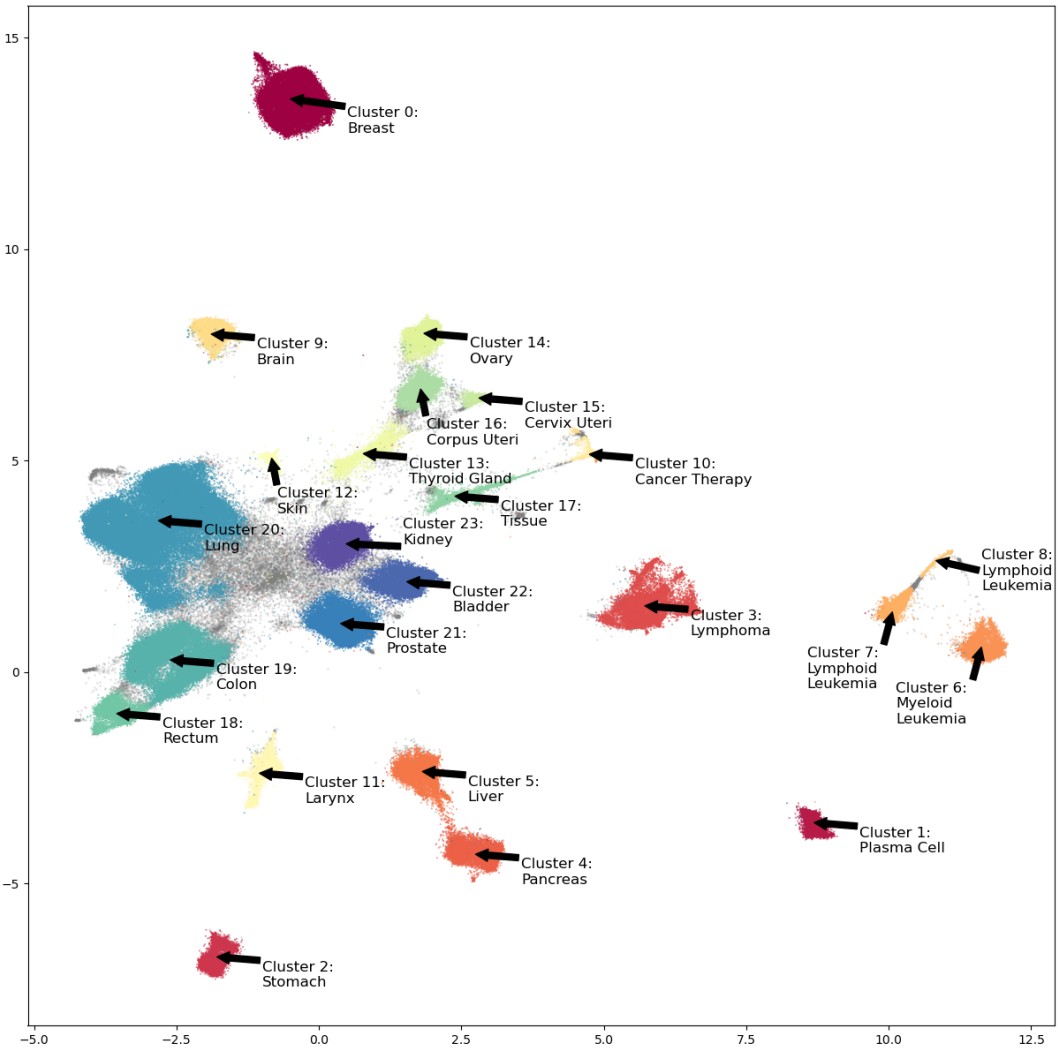

Figure 8: The unsupervised cluster assignments from HDBSCAN, visualized with a 2-dimensional UMAP projection. The grey points are patients not assigned to any cluster (10%). The labels indicate the most frequent diagnosis code of each cluster. Besides cluster 10, all label are neoplasms. On average, the most occurring cancer diagnosis within a cluster was present for 84% of the patients assigned to this cluster. The clusters are clearly separated spatially, indicating a distinct separation of the different cancer types and their representations within the model.

