# OpenReview forum: "ExBEHRT: Extended Transformer for Electronic Health Records"
_ICLR.cc/2023/Workshop/TML4H — ICLR 2023 Workshop TML4H Oral_

### Official Review · Reviewer_Z49m · 2023-02-28
**Leaning-to-Accept**

**Rating:** 8
**Confidence:** 5

**Review:**

The paper proposes a model, ExBEHRT, for Electronic Health Records (EHRs). ExBEHRT is an extended
version of BEHRT (BERT applied to Electronic Health Record data). ExBEHRT is different from BERT in
that its medical concepts are not concatenated into a single vector, but grouped based on concept
type. Experimental results have shown the effectiveness of the proposed approach in several
downstream tasks.

While the authors did provide some related work, it is very limited. They should cite more research
articles describing similar models and techniques. Ideally, there should be a separate section for
related work.

In the discussion section, authors made it seem as if the proposed approach for incorporating
features is the main contribution of the paper. The technique is interesting, but there is no
significant innovation in grouping by the concept type. The main contribution is the model itself,
which, according to the authors, was trained on "one of the largest EHR datasets from the USA."
While the dataset itself might not be accessible to researchers and engineers, it would be
interesting to see more statistics about the EHR data. Revealing the source of the data can also be
helpful.

Comparisons with XGBoost and BEHRT are both helpful, but not enough. It is also important to compare
the results of ExBEHRT with those of other transformer models in the healthcare domain such as
BioBERT or BioClinicalBERT (`Bio_ClinicalBERT` on Hugging Face).

The model training section needs more information. Since the authors used a (presumably) very large
dataset, more information about the training process including the optimizer, loss function, etc.
can be helpful. Was the model trained in the distributed manner? If so, what kind of distributed ML
paradigm was applied? The loss curves can be helpful as well. Information about the hardware they
used can be interesting to the audience.

While authors do hint on potential new directions of research, there is no separate section
dedicated to this - please add the future work section. There is no conclusion section either so
please add that as well.

The paper also needs to be revised (wording, phrasing, grammatical errors, etc).

Overall, I am leaning to accept, but the authors will need to address the points above.

Minor comments:

1. Page 1: In several sections (e.g., Introduction, Discussion, etc.), there are missing spaces
   in-between paragraphs. Put a newline or merge the two into one.

---

### Official Review · Reviewer_oSUe · 2023-03-01
**This study enhances Transformer models for Electronic Health Record data with multi-modal features**

**Rating:** 7
**Confidence:** 4

**Review:**

## Summary
This study enhances the performance of Transformer models for Electronic Health Record data by incorporating multi-modal features. The proposed method adds medical concepts separately and vertically instead of horizontally. The study demonstrates the significance of these features in several downstream applications.

## Strengths
[S1] The study addresses a significant problem in the field of electronic health records, which is an interesting and important area of research.

[S2] ExBEHRT's incorporation of additional features and interpretability allows for more informed and trustworthy decision-making in various downstream tasks of different diseases.

[S3] The extension is reasonable. Also, the approach of adding medical concepts separately and vertically is innovative and could lead to further advances in the field.

## Weakness
[W1] Table 1 mixes the dataset sizes (train, val, test) with the performance metrics (APS, AUROC, Precision), which can be confusing to interpret.

[W2] "compared to other, similar algorithms" -> "compared to other similar algorithms"
Also, it would be beneficial to include citations to several related works here.

[W3] The references section would be improved by including the venues (e.g., conference or journal names) for better identification and verification.

---

### Meta-Review · Area_Chair_zsRv · 2023-03-05

**Recommendation:** Accept (Poster)
**Confidence:** 5

**Metareview:**

The reviewers acknowledged the contributions of this paper and suggested accepting this paper. In the final version, please carefully consider the concerns raised by reviewers and further improve the paper.